# Non-Additive Effects of Environmental Factors on Growth and Physiology of Invasive *Solidago canadensis* and a Co-Occurring Native Species (*Artemisia argyi*)

**DOI:** 10.3390/plants12010128

**Published:** 2022-12-27

**Authors:** Bin Yang, Miaomiao Cui, Zhicong Dai, Jian Li, Haochen Yu, Xue Fan, Susan Rutherford, Daolin Du

**Affiliations:** Institute of Environment and Ecology, Academy of Environmental Health and Ecological Security, School of the Environment and Safety Engineering, Jiangsu University, Zhenjiang 212013, China

**Keywords:** additive effects, alien species, biological invasions, Canada goldenrod, global change, multi-factor effects

## Abstract

Changes in environmental factors, such as temperature and UV, have significant impacts on the growth and development of both native and invasive plant species. However, few studies examine the combined effects of warming and enhanced UV on plant growth and performance in invasive species. Here, we investigated single and combined effects of warming and UV radiation on growth, leaf functional and photosynthesis traits, and nutrient content (i.e., total organic carbon, nitrogen and phosphorous) of invasive *Solidago canadensis* and its co-occurring native species, *Artemisia argyi*, when grown in culture racks in the greenhouse. The species were grown in monoculture and together in a mixed community, with and without warming, and with and without increased UV in a full factorial design. We found that growth in *S. canadensis* and *A. argyi* were inhibited and more affected by warming than UV-B radiation. Additionally, there were both antagonistic and synergistic interactions between warming and UV-B on growth and performance in both species. Overall, our results suggested that *S. canadensis* was more tolerant to elevated temperatures and high UV radiation compared to the native species. Therefore, substantial increases in temperature and UV-B may favour invasive *S. canadensis* over native *A. argyi*. Research focusing on the effects of a wider range of temperatures and UV levels is required to improve our understanding of the responses of these two species to greater environmental variability and the impacts of climate change.

## 1. Introduction

Throughout the world, thousands of plant species have been moved from their native habitats and intentionally or accidentally introduced to new areas [1]. Some of these introduced plants have successfully naturalized and become invasive in the new range, replacing native plants, and causing harm to both the local environment and economy [2,3]. As such, plant invasions have increasingly become a hot research topic [4]. To explain why some species become successful invaders, a series of hypotheses have been put forward based on previous experimental research [5,6,7]. However, due to the complexity of biological invasions, these hypotheses are not necessarily applicable to all cases, with evidence supporting or contradicting each hypothesis (and some of the evidence is ambiguous) [7]. Moreover, with global environmental change, there are more variables, and therefore complexity, to consider when investigating plant invasions [8,9]. Consequently, research into plant invasions under global change is currently an important research direction and will continue to be so into the future [8].

It is generally expected that, under current and future global environmental changes, biological invasions will increase [9,10,11]. This includes increases in the total number of invasive species and more areas being affected by invasions [12,13,14]. The majority of studies focusing on the impacts of global change on plant invasions have explored the effects of a single environmental factor (e.g., temperature, precipitation, UV light) [10,11]. However, recent research has identified the importance of studying the interaction between multiple environmental factors when investigating the impacts of global changes on biological invasions [15,16,17]. In cases where the combined effects of two or more driving factors are equal to or have no significant impact compared to single effects, it is considered that there is an additive effect (i.e., no interaction between factors) on species [18,19]. In contrast, there may be a synergistic or antagonistic interaction between factors in cases where the combined effects have significant impacts on species [20,21]. Although, there is currently no unified definition of synergistic and antagonistic effects in the academic community [22,23,24], we follow the definition used by Crain et al. [25], which has been widely cited. Crain et al. [25] state that under scenarios where the stressors have negative effects when applied individually (e.g., stressor A reduces the response by ‘a’ and stressor B by ‘b’), then the cumulative effect under the condition of A + B is a change in the response compared to control levels and are as follows: additive (=a + b) and non-additive (<a + b, i.e., antagonistic; or >a + b, i.e., synergistic) [25]. Previous studies have shown that the response of plants to combined effects cannot be predicted by the combination of single factor effects [17,26,27]. Moreover, the interaction of multiple environmental change drivers may vary greatly due to the combination of test variables and drivers [21].

Over the past 200 years, human activities have warmed the climate at an unprecedented rate [28,29]. In the last 20 years, the average global surface temperature has increased by about 1 °C compared with the last century, and unless emissions of CO_2_ and other greenhouse gases are reduced in the next few decades, the global temperature is expected to increase by 1.5~2 °C by the end of this century [30]. Plant growth and development are very sensitive to climate warming [31], and global warming is expected to accelerate the spread of invasive species from warmer regions to higher latitudes and intensify its competitive advantage over native species [32,33]. Many habitats are already experiencing higher than average temperatures in response to warming and are less habitable for native plants adapted to colder conditions [34,35]. Temperature, which is an important climatic variable, has attracted extensive attention in the literature on the effects of global changes on plant invasions (e.g., Giejsztowt et al. [36], Gianoli and Molina-Montenegro [37], Liu et al. [38]).

While global temperatures are rising, several other environmental factors are also changing, including the amount of ultraviolet (UV) radiation reaching the earth’s surface, which is increasing mainly due to ozone depletion [30,39,40]. Although UV radiation is an important factor of global environmental change and can significantly impact plant communities [41], it is largely ignored in the study of plant invasions [42,43,44]. UV radiation can negatively affect plants by damaging their DNA, inhibiting photosynthesis (i.e., by destroying pigments involved in the photosynthetic apparatus), and changing hormone levels, thereby reducing plant growth [45,46]. Thus, when exposed to strong UV radiation, the biomass and height of plants will often decrease. Some studies have found that plants may increase their antioxidant capacity and accumulate flavonoids in response to small amounts of UV radiation in order to enhance their UV tolerance and ensure normal growth [47,48]. High UV radiation can also negatively impact plant communities, especially at high elevations and in the polar regions [49]. Given that different plant species are likely to vary in their sensitivity to UV levels, it is expected that species composition will change in response to increased UV, and interspecific relationships within the community will be affected [50]. In one study of 25 herbaceous species from Germany and New Zealand, Hock et al. [51] found that pre-adaptation to UV levels in the native range was a better predictor than functional traits of the success of these species in high UV environments in the invaded range. In another study, Hock et al. [43] revealed that eight exotic species displayed high phenotypic plasticity in response to different UV-B environments, which could be an important factor increasing the spread of alien species in the introduced range.

The effects of changes in temperature and UV radiation on plants are often studied separately, with only a few studies focusing on their combined effects [52]. Han et al. [53] found that warming not only inhibits the growth of *Picea asperata* seedlings but also enhances their antioxidant capacity, improving their resistance to UV radiation stress. As such, warming may reduce some of the negative effects of UV (and thus the interaction has antagonistic effects) [53]. However, it is currently unknown how co-occurring native and invasive plants will respond to a warm and UV-enhanced environment. In the current study, we compared the growth performance, leaf functional traits, photosynthetic traits, and nutrient content of the invasive species, *Solidago canadensis*, and its co-occurring native species, *Artemisia argyi*, to increases in two global environmental change factors: temperature and UV radiation (under single and combined treatments). We tested the following hypotheses: (1) increasing temperature alone will inhibit growth performance of both invasive *S. canadensis* and native *A. argyi*; (2) only increasing UV will inhibit growth and photosynthetic traits of *S. canadensis* and *A. argyi*; (3) there will be an antagonistic effect on plants between temperature and UV radiation; and (4) *S. canadensis* will be less affected by changes in temperature and UV compared to *A. argyi*.

## 2. Results

### 2.1. Growth Traits

Temperature had a significant impact on total biomass, leaf area, and leaf mass of invasive *S. canadensis*, while increased UV significantly affected leaf area, leaf mass, and root to shoot ratio in this species (*p* < 0.01, Table 1). Plant community had a significant effect on root-to-shoot ratio and leaf mass in *S. canadensis* (*p* < 0.01, Table 1). Our results also revealed that the interaction between temperature and UV significantly impacted leaf traits (leaf area and leaf mass, *p* < 0.05), as did the interaction between temperature, UV light, and plant community (*p* < 0.01, Table 1). When grown in monoculture, the total biomass, leaf area, and leaf mass of *S. canadensis* were inhibited by warming (traits decreased by 76.37%, 69.22%, and 39.11%, respectively, in the T- treatment compared to the control) and under warming and increased UV (traits reduced by 73.49%, 71.93%, and 41.36% in the U × T treatment, respectively, in Figure 1). In the mixed culture, warming also significantly inhibited total biomass, leaf area, and leaf mass in *S. canadensis* (which decreased by 76.37%, 78.40%, and 47.49%, respectively, in T- compared to the control). Root:shoot ratio and leaf mass in *S. canadensis* in the mixed community were significantly reduced when exposed to both warming and increased UV compared to the control (Figure 1). For *S. canadensis*, only leaf mass was significantly inhibited in response to increased UV alone in either the monoculture or mixed community (Figure 1).

For the native species, *A. argyi*, our results revealed that total biomass and leaf area were significantly affected by temperature (*p* < 0.01, Table 1). Leaf area was significantly impacted by UV light and the interaction between temperature and UV, while plant community significantly affected the root:shoot ratio in this species (*p* < 0.01, Table 1). The interaction between temperature, UV, and plant community significantly affected both leaf mass and leaf area (*p* < 0.01, Table 1). When grown in monoculture, total biomass and leaf area in *A. argyi* were significantly inhibited under the combined effects of warming and high UV (these traits decreased by 67.68% and 71.93%, respectively, in the U × T treatment compared to the control). In the mixed community, warming significantly reduced total biomass, leaf area, and leaf mass of *A. argyi* (which decreased by 80.58%, 82.16%, and 68.40%, respectively, compared to the control, Figure 1). Increased UV-inhibited leaf area of *A. argyi* in the mixed community (reduced by 66.08% in the U treatment, Figure 1). Although the mixed effects analysis indicated a significant effect of temperature on root:shoot ratio in *A. argyi* (Table 1), further exploration of the data using the nonparametric ANOVA revealed that this trait did not change significantly in response to single or combined effects of temperature and UV (*p* ≥ 0.05, Figure 1).

### 2.2. Photosynthesis Traits

We found that temperature significantly affected net photosynthetic rate, stomatal conductance, and Fv/Fm in *S. canadensis* (*p* < 0.05, Table 2). The interaction between temperature and UV had a significant effect on net photosynthetic rate and Fv/Fm (*p* < 0.05, Table 2). Stomatal conductance was significantly affected by plant community (*p* < 0.001) and the interactions between plant community and temperature (*p* < 0.001) and between community and UV (*p* = 0.014, Table 2). When grown in monoculture, warming significantly increased the net photosynthetic rate of *S. canadensis* (which was 196.53% higher in the T treatment than the control, Figure 2). In the invasive species monoculture, stomatal conductance significantly increased (by 24.04%) and Fv/Fm significantly decreased (by 11.33%) in response to both warming and high UV (compared to the control, Figure 2). In the mixed culture, we found that stomatal conductance in *S. canadensis* was significantly inhibited when exposed to high UV (reduced by 23.18% compared to the control, Figure 2).

For *A. argyi*, temperature significantly affected chlorophyll content (*p* < 0.05), while UV had a significant effect on stomatal conductance (*p* < 0.001, Table 2). The plant community significantly impacted stomatal conductance, chlorophyll content, and Fv/Fm (*p* < 0.05, Table 2). The interaction between temperature and UV significantly affected net photosynthetic rate, stomatal conductance, and chlorophyll content (*p* < 0.001, Table 2). While the mixed effects analysis indicated that net photosynthesis and Fv/Fm in *A. argyi* were significantly affected by temperature (Table 2), the nonparametric ANOVA revealed that these traits did not significantly vary among treatments (*p* ≥ 0.05, Figure 2). Stomatal conductance in *A. argyi* was significantly inhibited by warming (which reduced by 32.88% in T compared to the control) and increased UV (decreased by 42.80% in U compared to the control) but was not significantly affected by high temperature and UV (*p* = 0.938, Figure 2).

### 2.3. Plant Nutrient Ratios

In *S. canadensis*, the temperature significantly affected C:P and N:P, while UV had significant effects on all plant nutrient ratios (*p* < 0.001, Table 3). Plant community significantly impacted C:N and N:P in *S. canadensis* (*p* < 0.001), and C:N was significantly affected by all interactions between temperature, UV, and plant community (*p* < 0.05, Table 3). The interactions between temperature and UV and between the plant community and UV had significant effects on N:P in this species (*p* < 0.05, Table 3). When *S. canadensis* was grown in monoculture, its C:P content was significantly higher than the control group in response to warming (increased by 55.45% in T), and with increased temperature and UV (increased by 53.92% in U × T, Figure 3). In the mixed culture, C:N of *S. canadensis* decreased significantly under high UV (by 40.41%) and warmed with increased UV (Figure 3). Additionally, N:P of *S. canadensis* in the mixed community increased significantly in response to both warming and high UV (by 148.63%, Figure 3).

All plant nutrient ratios in *A. argyi* were significantly affected by temperature and UV (*p* < 0.05), while plant community had a significant effect on C:P (*p* = 0.003, Table 3). When grown in monoculture, C:P of *A. argyi* was significantly enhanced by warming and warming with high UV (compared to the control, C:P increased by 96.47% in T and 106.41% in U × T, Figure 3). In both the monoculture and mixed community, N:P of *A. argyi* showed a significant increase under the effects of warming and high UV (increased by 267.97%, Figure 3). The increase of UV, both alone and accompanied by warming, significantly reduced C:N in *A. argyi* when grown in the mixed community (decreased by 46.38% in U and 46.69% in U × T, Figure 3). In the mixed culture, C:P of *A. argyi* was significantly enhanced by temperature, both alone and with high UV (increased by 58.72% in T and 91.89% in U × T, Figure 3).

## 3. Discussion

In this study, we tested for the effects of temperature and UV light on a noxious invasive weed, *S. canadensis*, and a commonly co-occurring native species, *A. argyi*, when each species was grown in monoculture and together in a mixed community. Our findings revealed that growth and performance in both species were more significantly affected by warming than by UV radiation, with a higher number of traits being affected by temperature (8/11 and 9/11 traits in *S. canadensis* and *A. argyi*, respectively) compared to UV (6/11 traits for both species, Table 1, Table 2 and Table 3). Most growth traits in both species were significantly inhibited in response to warming and/or the combined effects of warming and UV when grown in a monoculture or mixed community (Figure 1). Although some photosynthetic traits in *S. canadensis* were enhanced in response to warming and/or warming with UV, stomatal conductance, and chlorophyll in *A. argyi* were significantly inhibited by the single or combined effects of warming and UV (Figure 2). Nutrient content ratios of native *A. argyi* were more affected by changes in temperature and UV light compared to *S. canadensis* (Figure 3). While performance in the invasive species was not vastly superior to the native species in this study, our findings suggest that *A. argyi* is nevertheless more impacted and changes in environmental drivers, particularly large increases in temperature and UV-B irradiation, may result in this species being at a disadvantage in the wild.

### 3.1. Single Effects of Temperature and UV on S. canadensis and A. argyi

Temperature affects nearly all aspects of plant growth and development, from seed germination to growth and senescence [54]. Our results revealed that warming suppresses the growth and performance of invasive *S. canadensis* and native *A. argyi* (Figure 1). As such, our first hypothesis that increasing temperature alone would inhibit growth in both species was supported. Our findings agree with a previous study where it was demonstrated that warming significantly inhibited growth in *S. canadensis* [55], as well as a meta-analysis of 177 publications, which found that increased temperature had a generally negative effect on plant root biomass [56]. In our study, warming reduced leaf area in both *S. canadensis* and *A. argyi* when each species was grown in monoculture and mixed culture (Figure 1). Gas exchange between leaves and the atmosphere is positively related to leaf area [57]. Here, although the net photosynthetic rate (per unit leaf area) increased in *S. canadensis* when exposed to warming (in monoculture), in *A. argyi* stomatal conductance and chlorophyll decreased under higher temperatures (Figure 2). Therefore, photosynthetic performance of *A. argyi* was inhibited in response to warming, which could be due to a reduction in leaf area (and thus photosynthetic area) under higher temperatures, resulting in an overall decline in plant productivity and biomass [58].

Previous studies have revealed that increases in UV radiation can damage the photosystem of plants, thereby affecting photosynthesis, growth, and development [59,60]. Compared with the robust structure of woody plants, herbaceous plants are often more responsive to environmental fluctuations [61,62]. When exposed to UV radiation, herbaceous plants mainly rely on UV absorption compounds in the vacuoles of epidermal cells as a defence mechanism, while in woody plants, such compounds exist in both the vacuoles and epidermal cell walls [63,64]. As such, herbaceous plants are more sensitive to UV radiation compared with woody species [61,65,66]. In this study, although increased UV significantly reduced stomatal conductance in both species and chlorophyll content of leaves in *A. argyi* in the monoculture, net photosynthetic rate, and maximum photochemical efficiency (PSII) were not affected by high UV alone (Figure 2). In terms of growth traits, the high UV radiation treatment only significantly inhibited leaf area in both species in the mixed community (Figure 1). Therefore, our second hypothesis that increasing UV alone would inhibit the growth and photosynthetic traits of *S. canadensis* and *A. argyi* was partially supported. One reason for this result could be that the maximum value of UV radiation applied in this study did not exceed the high levels of UV experienced by these species during the hottest times of the year. This is because our aim was to explore the cumulative effect of increasing UV radiation on these species over the longer term, rather than focusing on very high UV over the short term. Many studies have shown that although UV radiation destroys plant chlorophyll, the effect on photosynthesis may be relatively small [67,68,69]. Furthermore, moderate increases in UV over the long term may not have a significant effect on photosynthetic rate [70] because although UV radiation adversely affects plant tissues, it can also induce plants to produce stress-resistant secondary metabolites such as APX, POD, proline, and UV-B absorbing compounds [53]. Such compounds may enhance the resistance of plants to UV and other environmental stressors [71,72,73]. Our findings are consistent with another study, which found that increased UV radiation did not significantly affect net photosynthesis, organic matter accumulation, or biomass of plants growing in polar regions [74]. Nevertheless, many other studies have shown that UV has an inhibitory effect on plant growth and biomass accumulation, and different species likely vary in their sensitivity to enhanced UV radiation [67,74].

### 3.2. Combined Effects of Temperature and UV on S. canadensis and A. argyi

Previous studies suggest that increases in UV radiation and other stressors can have antagonistic interactions [52,75,76]. For example, a meta-analysis of 52 publications revealed that increases in plant defence responses under the combination of water stress and UV were less-than-additive, indicating that the impacts of increased droughts may be reduced by high UV light [77]. Similarly, the effects of heavy metals (i.e., cadmium) on algae were weakened by exposure to UV radiation [78,79], while the artificial chemical pollutant, dioron, was found to have an antagonistic interaction with UV on algal reproduction [79]. Of the few studies that have focused on the combined effects of warming and UV, most report a compensatory effect of temperature increases on UV-B-mediated plant growth inhibition [80]. For example, in *Helianthus annuus*, *Zea mays*, and *Picea asperata*, it was found that increases in UV reduced plant growth, chlorophyll content, and net photosynthesis rate, but these negative effects were lessened by high temperatures [53,80,81]. Other studies have shown that low levels of UV-B radiation can improve heat resistance in *Cucumis sativus* and conifer seedlings [82,83]. Therefore, there is likely to be an antagonistic interaction between increases in temperature and high UV-B on plants (i.e., each factor can weaken the effect of the other). In our study, we found that most traits in *S. canadensis* were significantly affected by the interaction between temperature and UV (6/11 traits, Table 1, Table 2 and Table 3). In contrast, less than half of the measured traits were significantly impacted by the interaction between temperature and UV in *A. argyi* (4/11, Table 1, Table 2 and Table 3). For *S. canadensis*, mostly growth traits were significantly inhibited in response to the combined effects of temperature and UV (e.g., total biomass, leaf area, root:shoot ratio, and leaf mass, Figure 1). In *A. argyi*, all nutrient ratios were affected by the warming and UV treatment (with a significant increase in C:P and N:P in both the monoculture and mixed culture, Figure 3). After performing calculations of relative interaction index (RIE) for all traits in both species (see Appendix A), we found there was an antagonistic interaction between temperature and UV for many traits in *A. argyi* (6/11 in the monoculture, 5/11 in the mixed culture, Appendix A). By contrast, for *S. canadensis*, fewer traits were affected by the antagonism between temperature and UV (3/11 in both the monoculture and mixed culture, Appendix A). As such, our third prediction that there would be an antagonistic effect between temperature and UV radiation on plants was only partly supported, with the native species being more affected than the invasive species. In *S. canadensis*, more traits were affected by a synergistic interaction between temperature and UV (5/11 in both the monoculture and mixed culture, Appendix A). Therefore, while *A. argyi* is more inhibited by warming than *S. canadensis*, the negative effects on many growth and performance traits in the native species were offset by antagonism between the two abiotic factors.

### 3.3. Performance of Invasive S. canadensis and Native A. argyi

Our findings revealed that more growth and performance traits in *A. argyi* were impacted by changes in temperature and UV than in *S. canadensis* (especially the nutrient ratios, Figure 3). Thus, our fourth hypothesis, that the invasive species would be less affected by the treatments than the native species, was supported. This result is consistent with many previous studies documenting that invasive plants have stronger growth performance than native plants [84,85,86]. Although growth and performance in *A. argyi* was more inhibited, the species was not highly disadvantaged compared with *S. canadensis*. Many traits in *S. canadensis* were significantly affected by high temperature and/or high UV (Table 1, Table 2 and Table 3), and the inhibitive effects of several traits in *A. argyi* were offset by antagonism between temperature and UV (Appendix A).

Numerous studies have confirmed that warming can enhance the performance of invasive plants over native plants [32]. Yet, in some situations, warming may reduce the invader’s advantage, particularly in areas where invasive and native plants grow together [87,88]. In our study, *A. argyi* and *S. canadensis* are from the warm temperate zone. However, the temperature in the greenhouse can reach above 40 °C, which makes the average temperature after warming far exceed the suitable temperature for these species, resulting in heat stress [55,89]. While the two species showed similar growth performance responses under warming, higher temperatures reduced chlorophyll content and maximum photochemical efficiency in *A. argyi*, the chlorophyll content and maximum photochemical efficiency of *S. canadensis* were mostly unaffected. Therefore, *S. canadensis* is more tolerant of warming [90] and could gain an advantage over *A. argyi* under higher temperatures and UV radiation over the longer term.

## 4. Materials and Methods

### 4.1. Study Species

*Solidago canadensis* is a perennial plant in the Asteraceae family, native to North America. It is distributed from North Dakota to Texas and Arizona in the United States, and from Nova Scotia to Ontario in Canada [91]. The species was first introduced to London (United Kingdom) around 1735 [92] and Shanghai (China) as an ornamental plant in 1935 [93]. Due to its rapid growth, developed rhizome, high seed yield, mode of dispersal (where it can spread widely via wind), and ability to colonize a wide range of environmental niches, *S. canadensis* has become a global invasive weed [94]. *Solidago canadensis* is distributed across a range of habitats, from mountains to rivers and marshes, forests and grass wastelands to farmland and urban areas, and can also be found along rivers and roadsides [95,96,97]. The total invasive range of *S. canadensis* in Lithuania alone is 1702 hectares [98], and in 2005 the species was estimated to have invaded 7787.97 hectares in Shanghai, China [99]. Like *S. canadensis*, *A. argyi* is a perennial plant in the Asteraceae and occupies a similar environmental niche in China, where it is native [89]. The distributions of *S. canadensis* and *A. argyi* in China overlap across a large area [89,100,101,102]. Given that these species are in the same plant family, have similar distributions in China, and are often co-occurring, *S. canadensis* and *A. argyi* were chosen as the invasive and native plants, respectively, for our study (Appendix A).

### 4.2. Experimental Design

The experiment was conducted in the greenhouse at Jiangsu University in the summer of 2021 (June to September). We collected seeds from a community of co-occurring *S. canadensis* and *A. argyi* in Nanshan (South Mountain) and Zhenjiang in Jiangsu Province, China (32.17° N, 119.45° E). Seeds were placed in square pots (15 × 15 × 12 cm) filled with soil collected from Zhenjiang, sand, and a nutrient mixture (in a ratio of 1:1:1). We allocated 20 pots to each of three cultures: native species monoculture (four seedlings of *A. argyi* were planted per pot), native and invasive mixed culture (two seedlings of *A. argyi* and two seedlings of *S. canadensis* were placed in each pot), and invasive species monoculture (four seedlings of *S. canadensis* were planted per pot). In our study, we used two types of temperature, ambient and warming: as well as two levels of UV, ambient and high. Six weeks after germination, each pot was assigned to one of four treatments in a full factorial design: ambient temperature and low UV (control group, CK), ambient temperature and high UV (U), warming and low UV (T), and high UV and warming (U × T). There were 5 pots (each pot comprising 4 individuals) per culture and per treatment. Therefore, there were a total of 240 plants in 60 pots.

Plants were randomly arranged inside four identical culture shelves (150 × 60 × 140 cm), each with one of the four treatments described above. The periphery of each culture rack was covered with a black shading net, which allowed plants to receive UV light from the top with the temperature being increased from the bottom. This design ensured the stability of O_2_ and CO_2_ levels inside the culture rack, minimizing the impact of uncontrolled microclimates on the pots. To better simulate the temperature changes in the natural environment, we did not set a fixed temperature but instead ensured that the ambient treatment in the experiment was as close to the natural local temperature (in Zhenjiang) as possible by ventilating the greenhouse. For the warming group, we set a treatment that would maintain a relative temperature difference of 2 degrees in real time as the environment changed (see Appendix A). To ensure that the heating in the culture rack was as uniform as possible in the warming treatment, two silicone rubber heating pads (60 × 55 cm, 500 W) were installed 50 cm below the basin of each culture rack. A temperature difference controller (Seasoon TN48, Shenzhen, China) was used to control the temperature so that it would start heating when the temperature of the warming group was less than 2 ℃ higher than that of the ambient group and would stop heating when the temperature was greater than 3 ℃. The temperature in each rack was recorded using a TH10R temperature and humidity recorder (Inste, Shenzhen, China). During the experiment, the temperature in each treatment was as follows: CK and U: 29.13 ± 0.25 ℃, and T and T × U: 31.36 ± 0.25 ℃ (mean ± SE, where the difference between CK and T was 2.23 ± 0.08 ℃ (Appendix A).

For our experiment, we used the ambient natural UV-B light in the greenhouse as the control treatment. We measured the natural light in the greenhouse at 50 cm above the soil surface of the pots using a UV-B radiometer (297 nm, Beijing Normal University Photoelectric Instrument Factory, China), and found that about 20% of the ambient UV-B was blocked. As such, the remaining UV-B light (21.541 ± 0.219 µw cm^−2^, mean ± SE) that reached inside the greenhouse was used as the basis for the low or ambient UV treatment in our study. Two UV-B lamps (TL 20W/12 RS SLV, Philips, Netherlands) were installed in each culture shelf about 100 cm above the soil surface in the pots. The lamps in two UV high treatment culture shelves were turned on for 14 h (6:00–20:00) every day during the experiment. In the UV high treatment, the UV intensity was increased by 10.936 ± 0.186 µw cm^−2^ compared to the control treatment (this was measured 50 cm above the soil surface layer in the pots using the radiometer after completely blocking the natural light).

The pot position in each culture rack was randomized once per week during the experiment. We harvested and recorded measurements from 240 individuals after 90 days of applying the treatments.

### 4.3. Data Collection

One fully expanded intact leaf in the centre of each plant was selected for measurements. Parameters such as photosystem efficiency, chlorophyll content, and net photosynthetic rate are useful indicators of a plant’s response to UV-B [103]. We measured the net photosynthetic rate (Pn) and stomatal conductance (Gs) of each leaf with a FS-3080H Plant Photosynthesis metre (Fansheng Technology, Shijiazhuang, China). The photosynthetic response of each plant was determined by measuring the chlorophyll fluorescence of each leaf using a PAR-FluorPen FP 110/D portable fluorometer (PSI, Brno, Czech Republic). To measure chlorophyll fluorescence, we prepared the leaf samples by applying the dark adaptation leaf clip to the leaves for 15 min. The meter was then used to measure the minimum fluorescence (Fo) and the maximal fluorescence intensity (Fm) using the OJIP mode. In the “OJIP” mode of the meter, the initial fluorescence intensity of 50 µs is Fo, and maximum photochemical efficiency (PSII) is calculated from the formula, Fv/Fm, where Fv = (Fm − Fo). The relative content of chlorophyll in each leaf was determined using a SPAD-502 PLUS chlorophyll meter (Konica Minolta, Tokyo, Japan). Additionally, for each leaf, the leaf area was measured using a leaf area meter (YMJ-CH, Tuopuyunnong, Hangzhou, China), and the fresh weight was determined using an analytical balance (BAS124S, Sartorius, Gottingen, Germany).

All plants were then harvested and dried at 60 °C for 72 h for total biomass and nutrient content measures. The biomass of all aboveground and belowground parts of each plant was determined using an analytical balance (BAS124S, Sartorius, Germany). Finally, the total organic carbon, nitrogen, and phosphorus of each plant after drying and grinding were determined using K_2_Cr_2_O_7_-H_2_SO_4_ oxidation, Kjeldahl nitrogen, and Molybdenum antimony resistance colorimetric (Agricultural Industry Standard of the PRC NYT 2017–2011) methods, respectively. The carbon-nitrogen (C:N), carbon-phosphorus (C:P), and nitrogen-phosphorus (N:P) ratios of each plant were calculated (Table 4).

### 4.4. Statistical Analysis

All statistical analyses were performed in R version 4.2.1 (R Core Team 2022). Plots of all measured traits were generated using the ggplot2 and ggsignif packages in R. Our preliminary exploration of the data using Levene’s and Shapiro-Wilks tests revealed that the normality and variance assumptions of the analysis of variance (ANOVA) were not met. Therefore, we used aligned rank transformation (ART) to implement the nonparametric factorial ANOVA. ART is a powerful and robust approach that is a nonparametric equivalent to the usual classical parameter analysis techniques and can be applied when the normality criterion is not met [104,105,106]. ART relies on one-step pre-processing to “align” the data before applying the averaged ranks, and then a nonparametric ANOVA can be used to test for treatment differences [107,108]. In our study, ART and nonparametric ANOVA were performed with the ARTool package version 0.11.1 [109] in R to evaluate the effects of temperature, UV light, plant species, and plant community (i.e., monoculture or mixed community) on plant growth and performance. ART was applied in a mixed-effect model using the art() function from *ARTool*. The anova() function (stats package version 4.2.0, R Core Team 2022) was then used to evaluate the significance of these effects, with plant number considered as a random effect. The contrast tests were implemented using the art.con() function (ARTool package) [107,110,111].

## 5. Conclusions

Our study showed that the growth of the invasive *S. canadensis* and a co-occurring native species, *A. argyi*, were inhibited under the influence of warming and increased UV-B, with both species being more affected by increased temperature. We found evidence of antagonistic interactions between temperature and UV on plant growth in both species, although more traits in *A. argyi* were affected by antagonism between the two factors than in *S. canadensis*. Although growth and performance traits in the two species were negatively impacted by single and combined effects of warming and UV, overall, *S. canadensis* was more tolerant than *A. argyi* in response to high temperature and UV-B irradiance. As such, under large increases in temperature and UV radiation, *S. canadensis* may be more resistant, and could gain an advantage over *A. argyi* in the wild. Future research should examine the growth performance of these two species when exposed to a wider range of temperature and UV radiation than explored in this study to assess the impacts of large-scale environmental changes.

## Figures and Tables

**Figure 1 plants-12-00128-f001:**
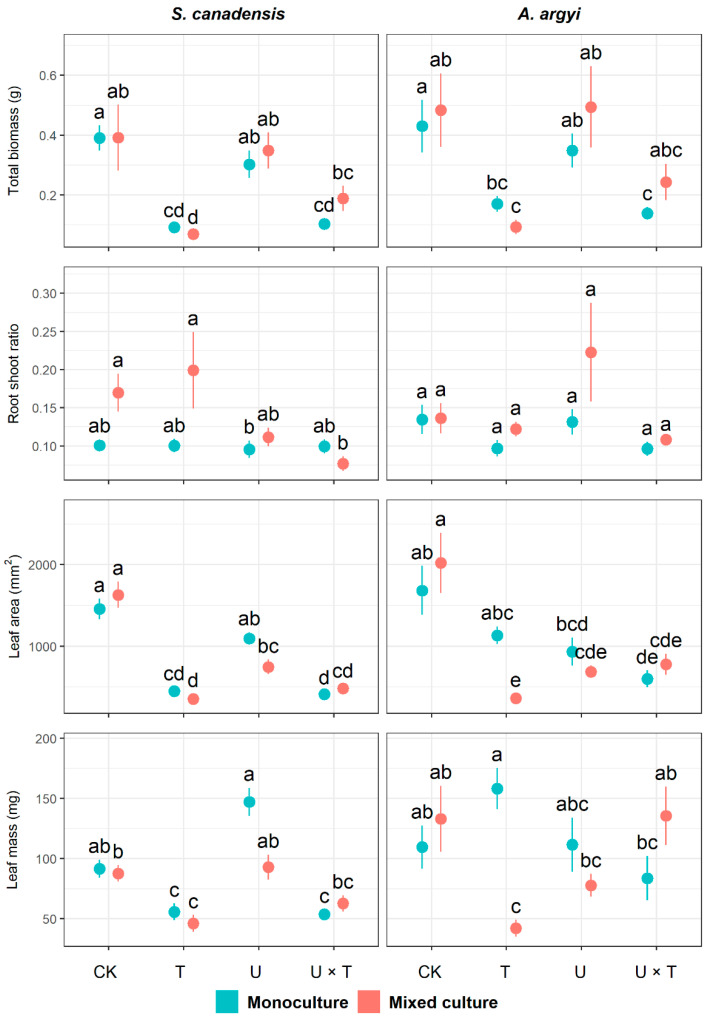
Effects of temperature and UV light on growth traits of the invasive *Solidago canadensis* and the co-occurring native species, *Artemisia argyi,* across plant communities (i.e., invasive species only, native species only, and mixed community). Each plant community was grown under four different treatments: ambient temperature and ambient UV light (control group, CK), warming and ambient UV light (T), ambient temperature and high UV (U), and high UV and warming (U × T). The solid data points represent the arithmetic mean, and the bars indicate the standard error (±SE). Significant differences (*p* < 0.05) among treatments are denoted by different letters above the data points.

**Figure 2 plants-12-00128-f002:**
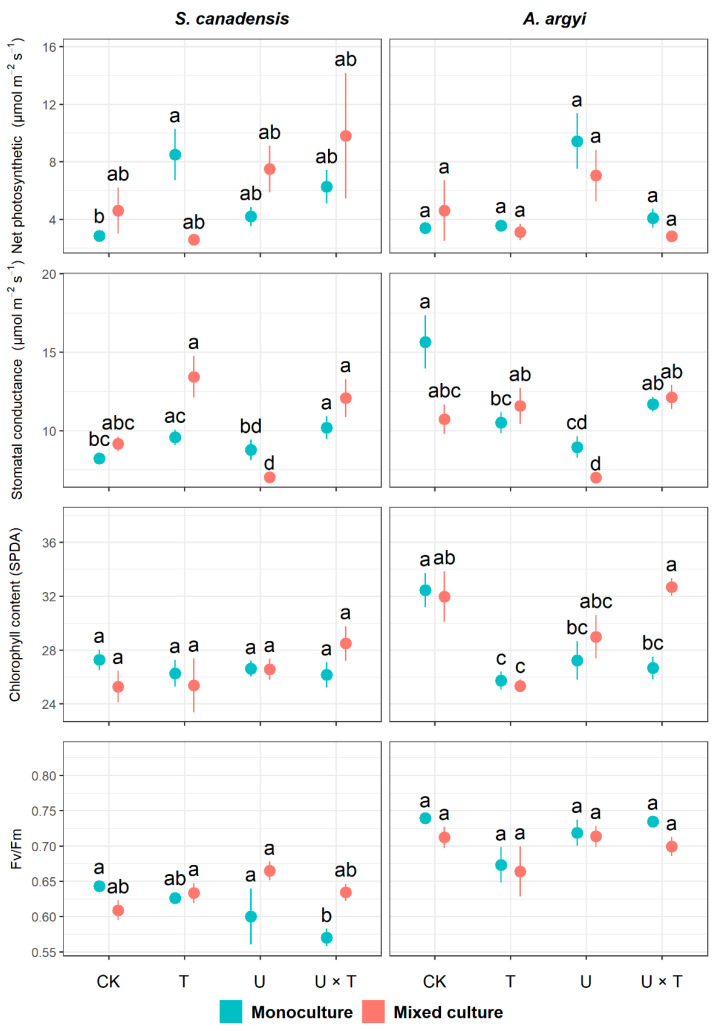
Effects of temperature and UV light on photosynthetic traits of the invasive *Solidago canadensis* and the co-occurring native species, *Artemisia argyi,* across plant communities (i.e., invasive species only, native species only, and mixed community). Each plant community was grown under four different treatments: ambient temperature and ambient UV (control group, CK), warming and ambient UV (T), ambient temperature and high UV (U), and high UV and warming (U × T). The solid data points represent the arithmetic mean, and the bars indicate the standard error (±SE). Significant differences (*p* < 0.05) among treatments are denoted by different letters above the data points.

**Figure 3 plants-12-00128-f003:**
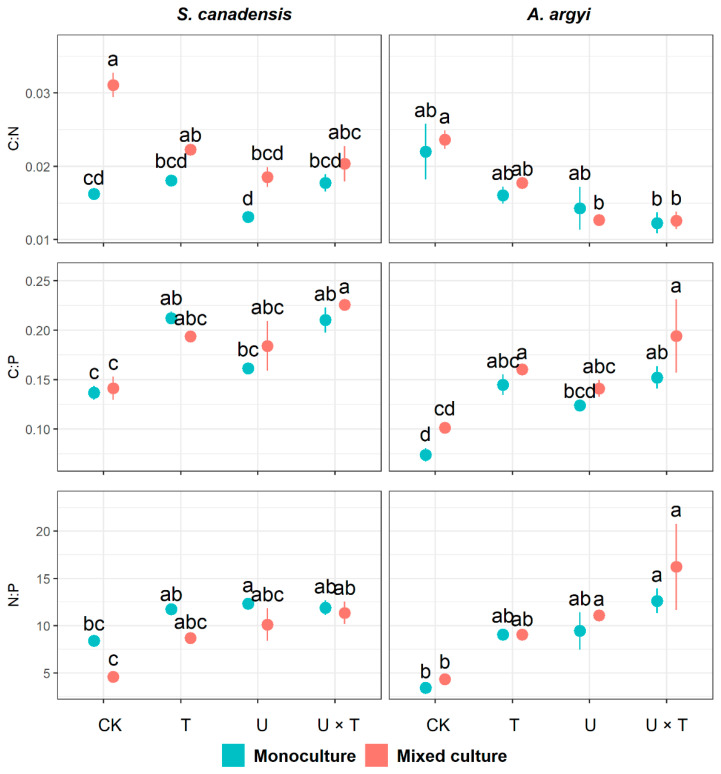
Effects of temperature and UV light on plant nutrient ratios in invasive *Solidago canadensis* and the co-occurring native species, *Artemisia argyi,* across plant communities (i.e., invasive species only, native species only and mixed community). Each plant community was grown under four different treatments: ambient temperature and ambient UV (control group, CK), warming and ambient UV (T), ambient temperature and high UV (U), and high UV and warming (U × T). The solid data points represent the arithmetic mean, and the bars indicate the standard error (±SE). Significant differences (*p* < 0.05) among treatments are denoted by different letters above the data points.

**Table 1 plants-12-00128-t001:** Mixed effects analysis of temperature (T), UV light (U), and plant community (C) on growth traits of invasive *Solidago canadensis* and native *Artemisia argyi* when grown in monoculture and together in a mixed community.

Source	*S. canadensis*				*A. argyi*			
	Total Biomass	Root: Shoot Ratio	Leaf Area	Leaf Mass	Total Miomass	Root: Shoot Ratio	Leaf Area	Leaf Mass
T	81.22 ***	0.13	159.74 ***	92.31 ***	26.70 ***	7.09 **	25.13 ***	0.00
U	0.41	8.57 **	26.30 ***	13.37 ***	0.05	0.29	21.50 ***	0.32
C	0.22	12.75 ***	0.55	8.24 **	0.97	8.02 **	2.09	2.09
U × T	2.46	0.00	36.24 ***	4.26 *	0.29	0.54	16.10 ***	0.95
T × C	0.05	0.75	0.17	6.71 *	0.96	0.48	1.00	0.20
U × C	3.39	14.35 ***	3.30	2.46	0.79	1.53	1.90	6.79 *
U × T × C	0.05	1.82	11.35 **	10.17 **	0.50	2.38	8.82 **	18.57 ***

Values provided in the table are F values. *, ** and *** indicate a significant difference at *p* < 0.05, *p* < 0.01 and *p* < 0.001, respectively.

**Table 2 plants-12-00128-t002:** Mixed effects analysis of temperature (T), UV light (U), and plant community on photosynthetic traits of invasive *Solidago canadensis* and native *Artemisia argyi* when grown in monoculture and together in a mixed community.

Source	*S. canadensis*				*A. argyi*			
	Net Photosynthetic Rate	Stomatal Conductance	Chlorophyll Content	Fv/Fm	Net Photosynthetic Rate	Stomatal Conductance	Chlorophyll Content	Fv/Fm
T	4.71 *	40.67 ***	0.00	10.82 **	8.93 **	0.27	7.25 **	5.10 *
U	0.86	0.96	0.10	1.80	19.39 ***	15.94 ***	0.76	2.21
C	0.02	13.86 ***	0.03	2.80	3.27	4.34 *	5.20 *	6.00 *
U × T	4.04 *	0.00	0.53	8.50 **	16.89 ***	46.01 ***	21.36 ***	3.82
T × C	2.87	15.83 ***	1.17	6.55 *	1.64	15.80 ***	3.16	0.03
U × C	5.19 *	6.20 *	2.77	8.32 **	0.21	2.65	8.40 **	0.29
U × T × C	0.42	0.90	0.18	0.03	0.52	4.09 *	1.66	0.09

Values provided in the table are F values. *, ** and *** indicate a significant difference at *p* < 0.05, *p* < 0.01 and *p* < 0.001, respectively. Fv/Fm represents primary light energy conversion efficiency of PSII.

**Table 3 plants-12-00128-t003:** Mixed effects analysis of temperature (T), UV light (U), and plant community (C) on plant nutrient ratios of invasive *Solidago canadensis* and native *Artemisia argyi* when grown in monoculture and together in a mixed community.

Source	*S. canadensis*			*A. argyi*		
	C:N	C:P	N:P	C:N	C:P	N:P
T	0.02	37.71 ***	14.22 **	8.73 **	27.44 ***	18.64 ***
U	26.16 ***	11.80 **	22.08 ***	31.50 ***	17.75 ***	20.88 ***
C	50.82 ***	0.81	21.84 ***	0.71	12.14 **	4.24
U × T	16.10 **	1.62	8.68 **	2.66	3.78	0.66
T × C	21.90 ***	0.35	1.55	0.02	0.06	0.02
U × C	11.20 **	3.55	5.41 *	1.64	0.01	2.18
U × T × C	5.76 *	0.72	0.00	0.71	1.30	0.38

The values provided in the table are F values. *, **, and *** indicate a significant difference at *p* < 0.05, *p* < 0.01, and *p* < 0.001, respectively. C:N, C:P, and N:P represent organic carbon to total nitrogen, total carbon to total phosphorous, and total nitrogen to total phosphorous, respectively.

**Table 4 plants-12-00128-t004:** Abbreviations, units, and methodology for the traits measured in this study.

	Trait	Units	Method
Growth	Aboveground biomass	g	The dry weight of the plant above the base
	Underground biomass	g	The dry weight of the plant under the base
	Total biomass	g	TB = AB + UB
	Root shoot ratio	%	R/S = UB/AB × 100%
	Leaf area	cm^2^	Leaf area meter
	Leaf mass	G	Leaf fresh weight
Photosynthesis	Net photosynthetic rate	μmol m^−2^ s^−1^	FS-3080H plant photosynthetic measurement system
	Stomatal conductance	μmol m^−2^ s^−1^	
	Chlorophyll content	SPAD	SPAD-502 PLUS chlorophyll meter
	Primary light energy conversion efficiency of PSII	-	PAR-FluorPen FP 110/D portable chlorophyll fluorescence tester (Fm − Fo)/Fm
Plant nutrient ratios	Plant carbon nitrogen ratio	-	CN = OC (Organic carbon)/TN (Total nitrogen)
	Plant carbon phosphorus ratio	-	CP = OC/TP (Total phosphorus)
	Plant nitrogen phosphorus ratio	-	NP = TN/TP

## Data Availability

All data from this study is reported in the manuscript and the Appendix A.

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
