# Peer review of "Non-Additive Effects of Environmental Factors on Growth and Physiology of Invasive Solidago canadensis and a Co-Occurring Native Species (Artemisia argyi)"

_plants, 2022, doi:10.3390/plants12010128_

Round 1

Reviewer 1 Report

Dear Authors,

reviewing the manuscript entitled: "Non-additive effects of environmental factors on growth and physiology of invasive Solidago canadensis and a co-occurring native species (Artemisia argyi)" by Bin Yang, Miaomiao Cui, Zhicong Dai, Jian Li , Haochen Yu, Xue Fan, Susan Rutherford and Daolin Du was a great pleasure for me.

The Authors investigated single and combined effects of warming and UV radiation on growth, leaf functional and photosynthesis traits, and nutrient content of invasive Solidago canadensis and its co-occurring native species, Artemisia argyi.

In the light of the observed and forecasted climate changes, mainly related to warming, the subject of research undertaken by the Authors is very interesting and, above all, innovative.

In my opinion, the research was done very well. And the manuscript that the Authors prepared is almost perfect. The aim and hypothesis of the research are surprisingly clearly formulated. Conclusions and future directions of research are well presented. The Introduction and Discussion chapters are a real pleasure to read. The results are well and legibly presented.

I just have a few minor remarks, mostly technical, that Authors should include in their manuscript:

1) Instead of (TxU) should be (U x T), e.g. in L. 157, 194, 223, 225, 231...

2) Table 1 and 2 - the first row of the table - the first letters of all words should be capitalized.

3) Titles of subsections - the first letters of all words should be capitalized.

4) Figure 2 - the description of the Y axis is illegible ("Net photosynthetic ..." and ""Stomatal conductance ...".

5) Paragraph L 352-L 363 - please format correctly, justify and remove italics.

6) Please verify the entire manuscript once again in terms of adapting it to the requirements of the template required by the Plants journal. Please pay particular attention to the References section.

In conclusion, I strongly encourage the Editors of the journal Plants to consider publishing this manuscript.

Author Response

Reviewer 1 : Review Report (Round 1)

Dear Authors,

reviewing the manuscript entitled: "Non-additive effects of environmental factors on growth and physiology of invasive Solidago canadensis and a co-occurring native species (Artemisia argyi)" by Bin Yang, Miaomiao Cui, Zhicong Dai, Jian Li , Haochen Yu, Xue Fan, Susan Rutherford and Daolin Du was a great pleasure for me.

No response necessary.

The Authors investigated single and combined effects of warming and UV radiation on growth, leaf functional and photosynthesis traits, and nutrient content of invasive Solidago canadensis and its co-occurring native species, Artemisia argyi.

No response necessary.

In the light of the observed and forecasted climate changes, mainly related to warming, the subject of research undertaken by the Authors is very interesting and, above all, innovative.

No response necessary.

In my opinion, the research was done very well. And the manuscript that the Authors prepared is almost perfect. The aim and hypothesis of the research are surprisingly clearly formulated. Conclusions and future directions of research are well presented. The Introduction and Discussion chapters are a real pleasure to read. The results are well and legibly presented.

No response necessary.

I just have a few minor remarks, mostly technical, that Authors should include in their manuscript:

No response necessary.

1) Instead of (TxU) should be (U x T), e.g. in L. 157, 194, 223, 225, 231...

Response: The "T×U" in the full text has been replaced with "U × T", and the corresponding notes and figure have also been modified synchronously. The specific modification involves L. 128, 142, 157, 195, 213, 221, 225, 227, 233, 399; Table 1, 2, 3; and Figure 1, 2, 3.

2) Table 1 and 2 - the first row of the table - the first letters of all words should be capitalized.

Response: The first letters of all words in the first row of Table 1, Table 2 and Table S2 have been changed to capital.

3) Titles of subsections - the first letters of all words should be capitalized.

Response: The titles of subsections have been modified.

4) Figure 2 - the description of the Y axis is illegible ("Net photosynthetic ..." and ""Stomatal conductance ...".

Response: The Y-axis of Figure 2 has been modified.

5) Paragraph L 352-L 363 - please format correctly, justify and remove italics.

Response: This paragraph has been adjusted to the correct style.

6) Please verify the entire manuscript once again in terms of adapting it to the requirements of the template required by the Plants journal. Please pay particular attention to the References section.

Response: We have proofread the manuscript and it is consistent with the requirements of the journal.

In conclusion, I strongly encourage the Editors of the journal Plants to consider publishing this manuscript.

No response necessary.

Reviewer 2 Report

This is a good, well-written and interesting paper. It describes results of physiological study on one of the most invasive plant species Solidago canadensis that is invasive not only in China but also in Europe. The results can be interesting for many readers.

I have only minor comments:

Lines 110-115: why synergistic effect of UV and temperature was not taken into account. Although authors wrote that “Previous studies have shown that the 61 response of plants to combined effects cannot be predicted by the combination of single factor effects” it is still not clear. Write more about it.

Figure 3: what is shown on figure? Arithmetic means, marginal means (emmeans package is being used along ARTool) or medians?

Lines 352-363: the text is in italics, change it into normal font.

Lines 377-378:  Only one sentence about native species is too few information. Write something more about the traits of the species. I suggest to prepare a table with comparison with plant traits of both species. It can be included in supplementary file. The key information would be in which habitats, plant communities these species co-occur. Is Ambrosia argyi being displaced by S. canadensis? Are there any studies about competition between the two species?

Lines 473-474: statistical analysis seems to be appropriate but what was random effect in mixed effect model? A pot?

Author Response

Reviewer 2: Review Report (Round 1)

This is a good, well-written and interesting paper. It describes results of physiological study on one of the most invasive plant species Solidago canadensis that is invasive not only in China but also in Europe. The results can be interesting for many readers.

No response necessary.

I have only minor comments:

No response necessary.

Lines 110-115: why synergistic effect of UV and temperature was not taken into account. Although authors wrote that “Previous studies have shown that the 61 response of plants to combined effects cannot be predicted by the combination of single factor effects” it is still not clear. Write more about it.

Response: In lines 100-110, we have listed that the response of common plants (without considering the origin) to the interaction of temperature and UV is mostly antagonistic (we have made this clearer in line 104-105). Following on from this previous research, in Hypothesis 3 (line 113), we speculate that the interaction effect of TU on native and invasive plants is also antagonistic. Therefore, synergistic effect of UV and temperature was not considered.

Figure 3: what is shown on figure? Arithmetic means, marginal means (emmeans package is being used along ARTool) or medians?

Response: The points in the figure are arithmetic mean values. The ARTool package is only used to calculate the difference between data, and the intermediate data analyzed by it is not used for drawing. It has been indicated in the and figure note of Figure 1~3.

Lines 352-363: the text is in italics, change it into normal font.

Response: This paragraph has been adjusted to the correct style.

Lines 377-378:  Only one sentence about native species is too few information. Write something more about the traits of the species. I suggest to prepare a table with comparison with plant traits of both species. It can be included in supplementary file. The key information would be in which habitats, plant communities these species co-occur. Is Ambrosia argyi being displaced by S. canadensis? Are there any studies about competition between the two species?

Response: The comparison table between the two plants is provided in the supplementary material.

Lines 473-474: statistical analysis seems to be appropriate but what was random effect in mixed effect model? A pot?

Response: The random effect in the mixed effect model is not pot, we used the plant number was considered for as random effect. Because in our analysis, we have been using each plant individual rather than pot as the unit to analyze data. At the same time, due to the existence of monoculture and mixed culture, the number of data in each pot is different, so we take the number of plants as the random effect in the mixed effect model. Relevant expressions have been added in line 437.
